# Genetic Algorithm-Driven Surface-Enhanced Raman Spectroscopy Substrate Optimization

**DOI:** 10.3390/nano11112905

**Published:** 2021-10-29

**Authors:** Buse Bilgin, Cenk Yanik, Hulya Torun, Mehmet Cengiz Onbasli

**Affiliations:** 1Electrical and Electrical Engineering, Graduate School of Sciences and Engineering, Koç University, Sarıyer, Istanbul 34450, Turkey; bebrem18@ku.edu.tr; 2Koç University Research Center for Translational Medicine, Koç University, Sarıyer, Istanbul 34450, Turkey; htorun18@ku.edu.tr; 3Sabanci University Nanotechnology Research and Application Center, SUNUM, Tuzla, Istanbul 34956, Turkey; cyanik@sabanciuniv.edu; 4Bio-Medical Sciences and Engineering, Graduate School of Sciences and Engineering, Koç University, Sarıyer, Istanbul 34450, Turkey

**Keywords:** surface-enhanced Raman spectroscopy, genetic algorithm, metasurface

## Abstract

Surface-enhanced Raman spectroscopy (SERS) is a highly sensitive and molecule-specific detection technique that uses surface plasmon resonances to enhance Raman scattering from analytes. In SERS system design, the substrates must have minimal or no background at the incident laser wavelength and large Raman signal enhancement via plasmonic confinement and grating modes over large areas (i.e., squared millimeters). These requirements impose many competing design constraints that make exhaustive parametric computational optimization of SERS substrates prohibitively time consuming. Here, we demonstrate a genetic-algorithm (GA)-based optimization method for SERS substrates to achieve strong electric field localization over wide areas for reconfigurable and programmable photonic SERS sensors. We analyzed the GA parameters and tuned them for SERS substrate optimization in detail. We experimentally validated the model results by fabricating the predicted nanostructures using electron beam lithography. The experimental Raman spectrum signal enhancements of the optimized SERS substrates validated the model predictions and enabled the generation of a detailed Raman profile of methylene blue fluorescence dye. The GA and its optimization shown here could pave the way for photonic chips and components with arbitrary design constraints, wavelength bands, and performance targets.

## 1. Introduction

Surface-enhanced Raman spectroscopy (SERS) is a highly sensitive and specific molecular detection technique based on nonlinear Raman scattering. SERS can deliver molecule-specific information on samples for many different analyses such as cancer biomarker detection [1], identification of bacteria [2], proteins [3], microRNAs [4], and DNAs [5]. The essential features of a good SERS molecular detection system are a high signal-to-noise ratio (SNR), Raman signal enhancement, and the reproducibility of the signal [6]. SERS substrates, which can increase the intensity of molecule-specific peaks with higher contrast, are preferred because they provide a higher SNR [7]. Besides the SNR, Raman signal enhancement is a critical parameter related to the performance of the system and is dramatically affected by the structure and the material of the SERS substrate on which the analyte molecules are imaged or probed [7]. Recent studies focused on increasing the SERS signal enhancement using different nanostructures. Camargo et al. used silver nanocubes and an experimentally measured enhancement factor (EF) of 2 × 10^7^ [8]. Elsayed et al. used silver and silicon NPs to improve the enhancement and showed that an EF of 10^5^ for silver nanospheres, 10^9^ for silicon nanowires, 2 × 10^9^ for the combination of silver nanospheres and silicon nanowires could be reached according to the electromagnetic simulations [9].

The comparison of the performance of SERS chips using only the EF is quite challenging, as there are multiple definitions of the EF in the literature [10]. In addition, the EF of different types of SERS structures (i.e., dynamic and static substrates) cannot be compared directly [11]. Furthermore, the EFs are generally calculated using the spectra collected from the hot-spot regions, and these regions are randomly distributed and may not be fabricated reproducibly [12]. A recent study demonstrated that the hot-spot regions of the SERS substrate contributes 24% of the overall SERS intensity [13]. Therefore, controlling the distribution of localized areas on metasurfaces is an important issue for SERS substrate design. The first method tries to test basic SERS nanostructures by changing their size; however, geometric scaling gives a limited improvement since the overall structure does not change much. Nonuniform structures could be an alternative for advanced SERS substrate design. Since there are infinitely many nonuniform geometries, an efficient design method to search for the solution set over arbitrary SERS constraints should be developed. Therefore, a topology optimization step is needed to optimize the structure of the SERS substrate for the desired Raman signal enhancement performance. While finite-difference time-domain (FDTD) models can be used for an exhaustive SERS substrate geometry optimization, such a systematic sweep becomes prohibitively time consuming in a realistic design due to the number of geometric and material degrees of freedom. Thus, more efficient design methods are essential.

Different topology optimization algorithms were presented in the literature [14,15,16]. To eliminate the prohibitive costs of FDTD-based geometric sweeps and development costs, increasing noise tolerance, to find the global optimum geometry [17], the genetic algorithm (GA) was preferred in this study for SERS substrate optimization. The GA is an adaptive heuristic search algorithm that imitates the process of natural selection in order to find the fittest offspring. It is an effective technique for nonlinear problems with multiple local solutions. The GA is becoming more commonly used in optics for the optimization of polarization rotators [18], integrated optical devices [19], and biosensors [20].

In this study, a GA was used to design a SERS substrate that has a controllable and homogeneously distributed E-field localization over the surface. In our optimization algorithm shown in Figure 1, we used FDTD models and nanostructure geometry revisions on a 2D periodic single-unit cell. The electric field (e-field) enhancement factors (EFs) were calculated using FDTD with 2D periodic boundary conditions, and the GA revised the unit cell nanopattern geometry at each iteration. First, the parameters of the algorithm were tuned to increase the optimization efficiency of the GA. After the optimal design had been selected, the nanostructure (NS) was patterned onto the silicon substrate with an electron beam (E-beam) lithography technique. The fabricated substrate was tested under the Raman spectroscopy in order to compare the simulation results with the experimental characterization.

## 2. Materials and Methods

### 2.1. SERS Substrate Optimization with the Genetic Algorithm

The GA is a heuristic search algorithm based on the natural selection process that mimics Darwin’s theory of evolution by encoding the solutions into matrices and crossing them with each other to produce the best solution based on their performances [21]. The algorithm is composed of five main parts: the generation of the solutions, the fitness function, the selection function, crossover, and mutation. Each solution is encoded into a matrix called an individual. Individuals could be binary or float matrices for different optimization strategies. Binary-coded individuals were used for the SERS substrate optimization where “1” represents the presence of gold and “0” represents the absence of gold in the selected area. A certain number of individuals were created randomly to generate an optimization set, which is called a population. The number of individuals making up a population is a user-specified parameter and needs to be optimized for different applications. In order to sort the individuals and select the best solution according to their fitness to the problem, a problem-specific fitness value has to be determined using a function returning the fitness success. Since the fitness value is the quantitative representation related to the probability of the selection of the individuals, choosing a proper fitness function is the most important part of the GA design. Previously, different fitness functions, such as the enhancement factor, refractive index change, and wavelength shifts, have been implemented [22], for different plasmonic topology optimization problems.

There are two different enhancement mechanisms for SERS that can be used as a fitness function: electromagnetic and chemical enhancement. Electromagnetic enhancement (EME) is a physical enhancement due to the local electromagnetic fields enhanced by the resonant excitation of plasmons, which is directly related to the size, shape, and material of the SERS substrate [23]. Chemical enhancement (CE) is related to the electronic polarizabilities of the analytes. The overall EF of a SERS substrate is the sum of its CE and EME; however, CE is generally orders of magnitude lower than EME. Therefore, the overall EF of SERS could be estimated using only EME, which could be calculated using the E-field distribution provided by FDTD analyses. The EME is approximated as [24]:(1)$$\;EF_{EME}\left(w_{0},w_{R},r_{m}\right)\approx {\left|\frac{E_{\mathrm{loc}}\left(w_{0},r_{m}\right)}{E_{0}\left(w_{0},r_{m}\right)}\right|}^{2}.{\left|\frac{E_{\mathrm{loc}}\left(w_{R},r_{m}\right)}{E_{0}\left(w_{R},r_{m}\right)}\right|}^{2}\approx {\left|\frac{E_{\mathrm{loc}}\left(w_{0},r_{m}\right)}{E_{0}\left(w_{0},r_{m}\right)}\right|}^{4}$$
where ω_0_ is the photon frequency, ω_R_ is the Raman scattered frequency, *E*_0_ is the incident electric field strength, *r*_*m*_ is the position, and *E*_loc_ is the local electric field strength. It is represented as the fourth power of the electromagnetic field, |*E*|^4^. The optimization process was investigated under the EF as a fitness function to find the optimal algorithm structure for SERS substrate design, which is that EME occurs on the high-localization regions called hot-spots.

The individuals were ranked according to their calculated fitness values. The ones with the highest fitness values were kept for the next iteration, which is called elitism. Then, individuals that would contribute to the creation of further generations were selected from the remaining individuals by a selection algorithm. The selected individuals were randomly crossed with each other using a crossover function to generate new individuals. In order to increase the diversity of the population and to prevent from the convergence to a local minimum point, some of the individuals were randomly modified to generate slightly changed random topologies, which is called mutation. The new population was generated after all the parts of the algorithm were completed, and again, the fitness values of the new population were calculated. This loop continued until our two termination criteria were reached: (1) the termination criterion was the change lower than 0.001% in the fitness value for more than 10 generations and (2) the iterations reaching 100 generations.

In this study, the NS unit cell consisted of 10 × 10 px^2^ with 50 nm per px, which means that each individual was represented by a 1 × 100 vector. Since recent studies have shown that symmetric SERS substrates produce more reproducible signals compared with asymmetric substrates [25], a 4-fold symmetry constraint was applied on the optimization; therefore, only 25 genes were used for the representation of an individual. First, the parameters of the GA were optimized for SERS substrate optimization, then the optimized parameters were used to design an optimal SERS structure. The GA was designed and optimized using MATLAB software, except the fitness value calculation. The generated individuals were sent to the Lumerical FDTD solver, and the E-field over the substrate was simulated to be used to calculate EME.

### 2.2. Electromagnetic Simulations

The E-field distribution was calculated using the Lumerical FDTD solver. The gold cuboids were located on the silicon substrate. The total-field scattered-field source was used with a 0° polarization angle. The override mesh was used at the interface between the gold and the silicon. The conformal variant 1 mesh refinement method was used due to the existence of the metal. Antisymmetric and symmetric boundary conditions were used in the x-axis and y-axis, respectively. The unit cell was periodically patterned in the x-axis and y-axis. A perfectly matched layer boundary condition was used in the z-axis. The E-field was calculated with the frequency domain profile monitor located above the gold surface.

### 2.3. SERS Substrate Fabrication

The optimized SERS substrate was fabricated in the Sabanci University Nanotechnology Research and Application Center (SUNUM) Cleanrooms. A bilayer process was performed to achieve an easy lift-off. Si wafers were spin-coated with 495 poly(methyl methacrylate) (PMMA) C2 at 4000 rpm (rotations per minute) and baked at 170 °C for 5 min. Then, the 950 poly(methyl methacrylate) (PMMA) C2 was coated at 4000 rpm and baked at 170 °C for 5 min. The resist thickness was 180 nm. Next, the wafer was exposed to a Raith EBPG5000 plusES 100 kV electron beam lithography system with a low/small spot size current (100 pA) and high-resolution parameters at a 750 μC cm^−2^ e-beam dose. After exposure, the wafer was developed in 1:3 (by volume) MIBK:IPA (MIBK: methyl isobutyl ketone; IPA: isopropanol) for 1 min and 1:1 (by volume) concentration MIBK:IPA for 10 s, respectively. The wafer was then dipped into IPA for 30 s to stop the development, rinsed with IPA, and blow-dried with nitrogen. To eliminate any PMMA residues, 7 s of oxygen plasma was performed at 50 W, a 20 sccm O_2_ flow rate, and a 37.5 mTorr chamber pressure. After development and plasma cleaning, 5 nm Cr/50 nm Au layers were thermally evaporated on the wafer. The wafer was dipped in acetone overnight for lift-off. The chips were ultrasonicated in a bath for a short time, rinsed with acetone and isopropanol, then blow-dried with nitrogen.

### 2.4. Raman Spectroscopy Analysis

The SERS chips were analyzed with the methylene blue M9140 (Sigma-Aldrich, St. Louis, MO, USA) fluorescent dye. The 0.25 g of the dye powder was mixed with 50 mL double-distilled water and dropped onto the SERS substrate. The SERS chip was analyzed using with a 50× objective (Leica 50×/0.75), a 633 nm laser wavelength, 100% laser power (source power of 18 mW), 1 s of exposure, one accumulation, and cosmic ray removal using the Renishaw InVia Raman Microscope. The optimized structure was also compared with an industrial Raman substrate J13856-01 (Hamamatsu Photonics, Hamamatsu, Japan) under the same measurement parameters.

## 3. Results

### 3.1. Tuning the Parameters of the Genetic Algorithm

To optimize the GA, six substantial parameters needed to be determined: population, fitness, selection and crossover functions, elitism, and mutation ratios. Parameter tuning is an important step in the GA optimization as it can affect the convergence speed of the algorithm and the probability of reaching the global optimum. Except for the fitness function, the other parameters are the parameters that determine the structure of the GA. The fitness function quantifies the figure of merit of each structure. In Figure 2 and Figure 3, all curves shown reached the stopping criteria, although not all cases completed the same number of generations.

In this section, we present our refinement for the key genetic algorithm parameters. The experimental demonstration of the enhancement factor refinement over the generations that converged to the best performance would be the best unambiguous proof that the genetic algorithm works as an advantageous optimization technique. However, fabricating the semi-optimized structures pose significant lithography and lift-off challenges and may prevent detailed testing. Nevertheless, the state-of-the-art electromagnetic FDTD simulation tools can provide experimentally accurate results that can be validated quantitatively by scanning near-field optical microscopy techniques.

#### 3.1.1. Population

The population consists of two important parts: the initial population used to initiate the optimization and the size of the population. The initial population is a critical parameter since the future generations will be mostly composed of their offspring. There are two main strategies to generate the initial population: predefined or random. In predefined population generation, all or some of the individuals are included manually in the initial population. This method is suitable when the algorithm is used to reach a topology that exceeds the performance of a particular topology or when a powerful set of individuals in terms of performance is known in advance. If these conditions are not met, manually generating the population may adversely affect the performance of the algorithm. In order to see the effect of it, two different initial populations were generated and optimized with the same GA parameters. The first one was composed of randomly created individuals, and the second one was included a manually generated SERS substrate that was an unpatterned gold thin film structure. The average distances of the individuals in a population for each generation are shown in Figure 2A. Although a large part of the population was created randomly even in the predefined initial population, it was observed that the variation of individuals created in the predefined population generation case did not increase, but rather decreased. The main reason for this is that we manually intervened in the diversity of the seed population. Contrary to this case, if there were strong designs that we knew in advance, this could lead the simulation to converge faster. Therefore, how to use the initial population may yield different results depending on the problem. If there is a set of acceptable solutions regarding the structure to be optimized in advance, using a predefined initial design could improve the convergence speed. Otherwise, randomly generated individuals should be used to increase the diversity of the solution set.

Each population consists of a determined number of individuals. The individuals that will form the next generation are selected from among the ones in the previous population. However, an increase in the population size also causes the duration of the simulation to be prolonged. Since the E-field distributions calculated with the Lumerical FDTD solver were used to analyze the performance of individuals for SERS optimization, the fitness value calculation of an individual lasted approximately 30 s. This means that increasing the population size from 20 to 100 means increasing the optimization time from 8 h to 41 h for a GA limited to 50 generations. In addition to the simulation time drawback, the population size also affects the algorithm’s performance; therefore, SERS surfaces were separately optimized with three different sizes to analyze the effect of population size on optimization success: 20, 50, and 100. The best individual obtained after each generation for each optimization is shown in Figure 2B. It was observed that the population consisting of 20 individuals was not sufficient for effective optimization; therefore, it could not create enough variation and converged to the local maximum point. Although there was no critical difference between 50 and 100, it was observed that a set of 50 individuals gave better results. Increasing the population size too much both increases the time and may prevent the algorithm from converging to an optimal solution over the generations; therefore, the population size of 50 was chosen as the ideal population size. However, even if this number is used with the same fitness function and problem optimization, it may vary according to the size and structure of individuals; therefore, when a change is made in the algorithm in this manner, this parameter must be re-optimized.

#### 3.1.2. Selection Function

The selection function determines the individuals that will be used for the creation of the next generation regarding the fitness values of the individuals. The most common selection functions used in GA optimizations are the roulette wheel and tournament selection functions. The roulette wheel function selects an individual with a probability proportional to its fitness value [26]. The tournament function samples *n* number of individuals randomly and then selects the best individual among them [27]. The number *n* is determined by the user, and it dramatically effects the selection performance. Tournament selection with different *n* numbers and the roulette function were compared, and the achieved fitness values are shown in Figure 2C. The analyses proved that the *n* has an important effect on the optimization; therefore, the *n* has to be optimized in advance. Although the algorithm took longer to converge when the roulette function was used, it had the advantage that the roulette function had no parameters to optimize. While the roulette function caused the algorithm to converge over a longer time, it accelerated the algorithm tuning since it had no parameters to optimize. On the other hand, while the tournament function converged to the optimum result faster, it was necessary to spend a certain amount of time to optimize the parameter of the function. This trade-off may cause different functions to be preferred for solving different problems. The roulette wheel selection is suggested for a problem-specific SERS substrate optimization to reduce the complexity since the optimization process is already a complex process involving calculating the E-field distribution, modeling the EF, and optimizing with the GA. However, if a GA platform is desired to be developed for SERS optimization, the tournament function could be preferred since the optimization time of each substrate will become the most crucial parameter.

#### 3.1.3. Crossover Function

After the selection function determines the individuals that are responsible for the next generation, the crossover function combines two of them and creates their offspring. The combination strategy of the two selected matrices is a significant factor. There are two main methods that are widely used in the GA: single-point and two-point crossover functions. The single-point crossover function divides the matrices at a randomly selected single point and combines the divided matrices to create two new individuals. The two-point crossover function works on the same principle, but divides the matrix by two points, not one [28]. For this reason, the diversity of individuals created with the two-point crossover function is greater. To test this hypothesis, SERS substrates were separately optimized using with the two different crossover functions. The obtained fitness values are given in Figure 3A. It was observed that the optimization made with the two-point crossover function gave better results than one-point crossover function.

After the crossover function is determined, one should determine what percentage of the individuals created with the crossover function will constitute the new generation. The remaining individuals will be generated by the mutation; therefore, the crossover rate has an effect on both the determination of the true optimization direction and the increase in the diversity of the population to reach the global maximum point. In order to test the crossover rate, three different rates were used to optimize the SERS substrate, and the obtained fitness values are shown in Figure 3B. When the crossover rate was low, the GA could not determine the correct optimization path and converged to a local minimum; however, when it was increased too much, the improvements on the fitness value brought by the variation obtained by mutation could not be achieved.

#### 3.1.4. Elitism and Mutation Ratio

The last optimization parameters are the elitism and the mutation rates. The elitism rate determines how many individuals will be passed on directly to the next generation [29]. The mutation rate, on the other hand, determines what percentage of the genes that make up individuals will mutate [30]. If a large ratio is chosen as the elitism rate, the convergence efficiency of the GA may decrease as the number of individuals created by crossing will decrease in the new generations. Just as elitism, the optimal value of the mutation rate should be determined. In order to find the optimal numbers for SERS substrate optimizations, three different rates for each were used. The achieved fitness values are shown in Figure 3. It was seen that if a large mutation rate was chosen, the optimization time of the algorithm would take longer (or even, the GA may diverge if elitism is not implemented), and if a very low value was selected, it would converge to the local minimum point because of the insufficient variation in the further generations.

## 4. SERS Substrate Optimization

After all the parameters of the GA were analyzed, the optimal parameters were selected for SERS substrate optimization: random initial population, population size of 50, roulette wheel selection function, two-point crossover function with the ratio of 0.8, elitism of 1, and mutation rate of 0.15. In order to show that the GA is a suitable method for SERS substrate optimization, SERS surface design was made with the optimized algorithm. One of the most important performance parameters for SERS surfaces is the reproducibility of the Raman signal. In order to achieve this, SERS surfaces that provide localization in large areas should be preferred instead of structures that provide very intense E-field localization in narrow areas on the surface. For this reason, it is important how the E-field vector obtained by FDTD simulations is used in a fitness function. In this study, two different fitness functions were tested and their performances compared. The first function used was the hot-spot average EF function, which is the average E-field intensity collected from the hot-spot regions over the surface. The other function, overall average EF function, gives the mean of the E-field intensities over the substrate. The optimization results, the obtained structures, and their E-field profiles are shown in Figure 4.

The use of the average hot-spot EF allowed the algorithm to be optimized to the surface with a higher performance. The mean operation might unintentionally prevent the formation of highly localized areas. For this reason, the structure obtained with the hot-spot EF was fabricated for the experimental tests of the SERS surfaces. It was seen that the field was more localized on the sharp points and gaps, which is commonly observed in many previous SERS nanostructure studies. A highly enhanced E-field was obtained due to the nonuniform structures composed of multiple sharp points and small gaps between them. It was seen that the optimized structure did not localize the E-field onto the small sections; instead, the E-field was distributed over the NS, which increased the reproducibility and detection capabilities in sensing applications.

## 5. Experimental Analysis of the Optimized SERS Substrate

The optimized design was fabricated using an e-beam lithography. The fabrication parameters are explicitly given in the Materials and Methods Section. In order to prevent the unintended lift-off of the patterned areas, a 5 nm chromium adhesion layer was added between the gold layer and the silicon substrate. A scanning electron microscope image of the fabricated optimized geometry is given in Figure 5. The unit cells with 50 nm pixels were fabricated properly and patterned over the substrate with a 100 nm unit cell spacing.

The performance of the optimized SERS surface was analyzed with an organic dye, methylene blue (MB). The MB solution was dropped onto the silicon, plain gold, and SERS surfaces, and their Raman spectra were measured. The collected Raman spectra are shown in Figure 6. The measured Raman peaks and their band assignments are given in Table 1. The spectra of silicon was used as a background signal since the SERS surface was fabricated onto a silicon wafer, and the plain gold surface was analyzed for performance analysis. A broad peak centered at 960 cm^−1^ occurred due to the transverse optical phonons of silicon [31]. A decrease in the intensity of the silicon-specific peak in the Raman spectrum of SERS substrate was observed due to the presence of gold at the surface. The spectrum of the silicon substrate included just one peak related with the MB at 1626 cm^−1^, which was attributed to the C-C ring stretch. Since the displacement is generally bigger on the ring systems in heavy molecules, the presence of the ring stretching peak on the silicon substrate’s spectrum was expected [32]. This peak was also visible in the MB analysis on the plain gold surface. Besides, there were other weak peaks visible in the plain gold measurements attributed to the in-plane and out-of-plane bending of C-H and the symmetric and asymmetric stretch of C-N. However, their intensities were very low compared to the peaks obtained in the SERS analysis. There were three peaks that could only be detected during the analysis with the optimized SERS substrate: 1033 cm^−1^, 1299 cm^−1^, and 1331 cm^−1^. Since the deformation modes generally do not cause the creation of strong polarization changes, the Raman signals generated by them are generally weak [32]. The reason why these peaks could not be detected using a silicon or a plain gold substrate might be their weak Raman signal tendency. The optimized structure was tested also with an industrial SERS chip (J13856-01), and the obtained spectrum is shown in Figure 6. The optimized SERS substrate had a good performance comparable to the industrial SERS chip. As a result of these analyses, it was seen that the optimization of SERS substrates with the GA allowed a SERS system design with a comparable performance to the current SERS substrates and in which the distribution of hot-spots on the surface could be controlled.

The experimental EF was calculated with the method used in the previous studies [10,34,35,36,37]. The EF was calculated using the following equation:(2)$$\;EF=\frac{I_{SERS}}{C_{SERS}}\times \frac{C_{0}}{I_{0}}$$
where $I_{SERS}$ and *I*_0_ are the intensities of the peak at 1626 cm^−1^ measured from the SERS substrate and the reference, respectively. Similarly, $C_{SERS}$ and *C*_0_ are the concentrations of MB on to the substrates. Using Equation (Equation 2), the EFs of the SERS were obtained as 8.8 × 10^6^ and 6.3 × 10^6^ compared with the silicon and plain gold substrates, respectively. Since the assumptions made for the EF calculations were hardly fulfilled because of the surface roughness and defects, the EF was not used as a substantial performance criterion, but as a reference parameter that could give a clue about the performance of the SERS substrate. According to our findings, the GA could be utilized as an optimization tool for sensitive, repeatable, stable, and successful SERS substrate designs. These findings could pave the way for an application-specific SERS substrate development platform based on topology optimization algorithms.

## 6. Discussion

The most substantial part of the study was the reconfigurability of the algorithm for different applications. The optimization of integrated photonics is becoming more advantageous and attractive since photonic system performance could be dramatically improved. Table 2 shows selected previous research on the photonic optimization field. Neural networks have been used for the optimization of core–shell nanoparticles [38]; however, the complex network structure and network design make the optimization process complicated. Photonic devices optimized with the GA work in the long-wavelength range [39,40,41,42]. Moreover, the GA has been implemented for the optimization of a surface-plasmon biosensor [20]. One can observe that powerful designs could be obtained for different applications using the GA optimization technique. In addition to the GA, deep learning such as the generative adversarial network (GAN) is also used for optimizing the system [43,44]. Although the results of the GAN-based optimizations are promising, the performance of the network is directly related to the training set, which is generated manually. It is not trivial to generate the training data for the discriminator and design a system to evaluate the performance of both the discriminator and the generator. Therefore, if there an application-specific single design is needed, the GAN is unnecessary and more costly than the GA.

Complicated structures could dramatically improve the performance of the simple spherical SERS NSs that are frequently used for research. The power of topology optimization techniques is in their accelerated screening of high-figure-of-merit nanostructures out of many different geometries. In our case, each unit cell consisted of 10 × 10 (100) px, which can be reduced to twenty-five independent pixels due to the four-fold symmetry. Since each pixel can be a void or gold, there are 2^25^ different geometries, whose electromagnetic field profiles are prohibitively expensive (2^25^× 30 s = about 32 y) to calculate by brute force modeling. This solution space becomes even larger when considering different pixel types, materials, and system configurations. Using our GA, we started with a population of geometries that span a large entropy space, which helped us narrow down the SERS nanostructures with a near-global optimum figure of merit within only 1500 calculations.

Our methodology could also be used for different initial structures other than cuboids, which may improve the results. Researchers have used different unit cells for the SERS substrate design, such as spheres [45], nanocones [46], triangles [47], and nanostars [48]. SERS surfaces can be re-optimized by using these structures instead of cuboids. Besides changing the pixel shape, the substrate material might also vary. Gold, silver, and copper are metals commonly used in the visible and near-infrared wavelength ranges. In addition, alkali metals, semiconductors, and 2D materials such as graphene have also been used for different SERS substrate designs [49]. Since the GA can also be used for multi-objective optimization, material type can also be included as a parameter in the topology optimization. In addition, SERS substrate optimization can be performed for a single wavelength, as well as for a wavelength range. By using multi-objective GA, structures with the maximum and the minimum EF at certain wavelengths can be designed, making different areas of the SERS surface sensitive to different wavelengths. In this way, complex SERS substrates can be designed that can simultaneously detect analytes tagged with different Raman markers. Changing the fitness function could also improve the performance in other studies.

## 7. Conclusions

In this paper, the GA was used for the optimization of SERS substrates. This study showed that the optimization of SERS substrates improved the performance drastically and enabled the development of a SERS-based detection that might be used in analytical chemistry and potentially for future clinical applications. The binary-coded GA was modified for SERS optimization using the electromagnetic field distribution profile of a unit cell of the SERS substrate as a fitness value. The GA parameters were tuned in detail, and their effects on the optimization performance were discussed. This detailed optimization paves the way for the acceleration of the algorithm tuning to be performed if the GA would be used in the optimization of different SERS structures. This demonstrates that the GA is a powerful tool to design an application-specific SERS substrate that has a homogeneous localization distribution. To verify the simulation results with experimental data, the optimized design was fabricated with e-beam lithography and tested under Raman spectroscopy. An enhancement factor of >10^6^ according to the background was achieved with the fabricated substrate. This algorithm could be applied for other applications in SERS substrate design that require multi-objective optimization. 

## Figures and Tables

**Figure 1 nanomaterials-11-02905-f001:**
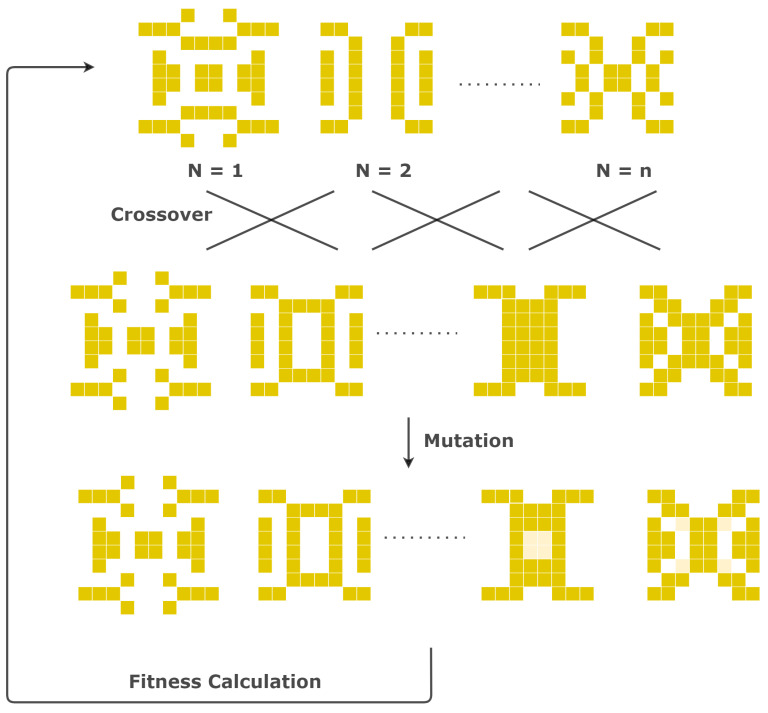
The workflow of one generation of the genetic algorithm (GA). The population number *n* is determined by the user. The SERS substrates were represented with binary-coded matrices: if it is 1 (or 0), the corresponding pixel is filled with metal (or air). The fitness values are the electromagnetic field enhancement factors and are calculated using the Lumerical FDTD solver.

**Figure 2 nanomaterials-11-02905-f002:**
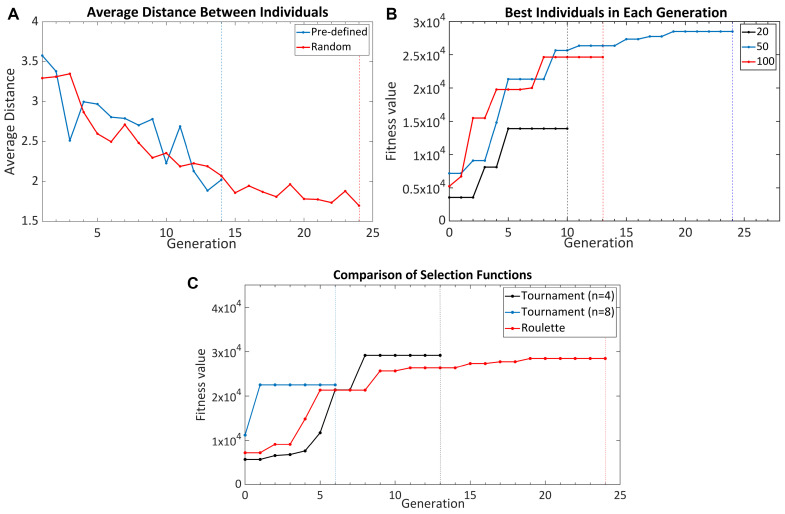
(**A**) The average distance between the individuals in each generation for different initial population functions. Since generating individuals randomly increases the population diversity, it is the recommended method when there is no prior information about the structure. (**B**) The fitness values obtained by GAs with different population sizes. According to the obtained results, the population size of 50 is the most suitable parameter for the SERS substrate optimization. (**C**) The fitness values obtained by GAs with different selection functions. Since the tournament function is affected by its parameters of *n*, the roulette wheel selection function is used for SERS substrate optimization.

**Figure 3 nanomaterials-11-02905-f003:**
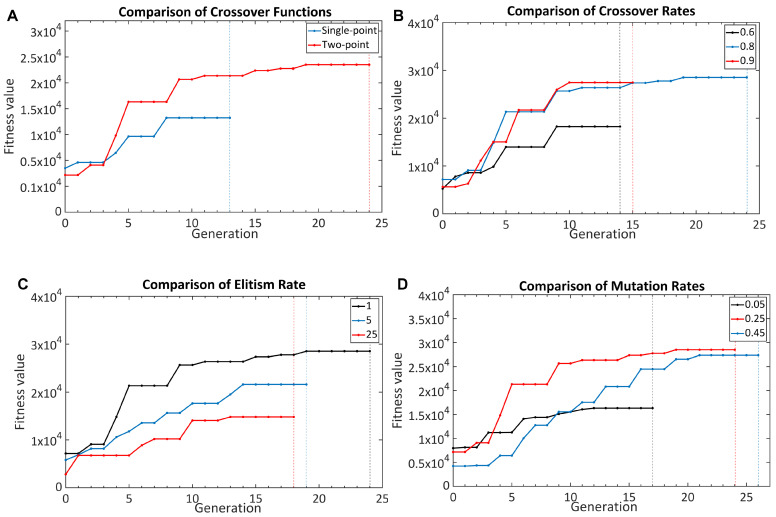
(**A**) Comparison of two different crossover functions. The two-point crossover function improves the performance of the GA due to the increased variety. (**B**) The effect of the crossover rate on the optimization performance. The crossover rate of 0.8 was set as the optimal parameter according to the obtained fitness values. (**C**) The effect of the elitism rates on the GA optimization. The optimal parameter for elitism is 1. (**D**) The effect of different mutation rates on the GA optimization. The mutation rate should be selected between 0.05 and 0.25.

**Figure 4 nanomaterials-11-02905-f004:**
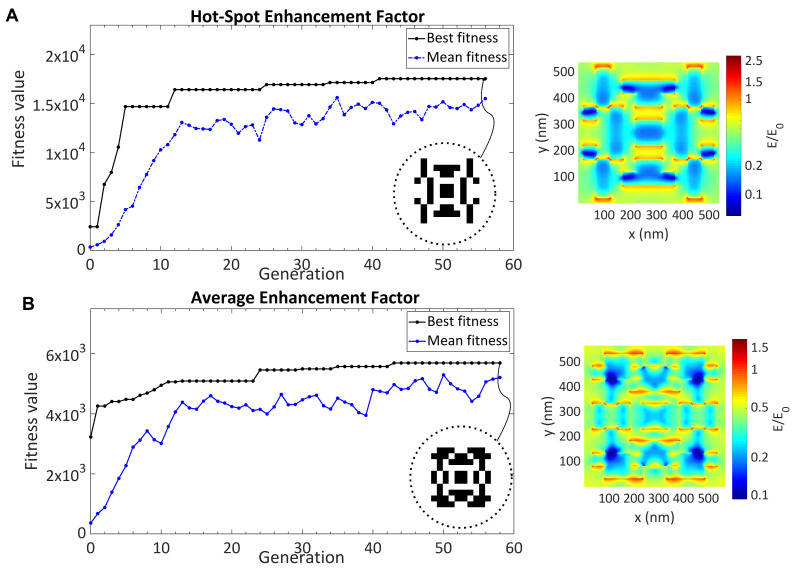
The optimization of the surface-enhanced Raman spectroscopy (SERS) substrate with the genetic algorithm (GA). (**A**) Optimization results using the hot-spot enhancement factor. (**B**) Optimization results using the average enhancement factor.

**Figure 5 nanomaterials-11-02905-f005:**
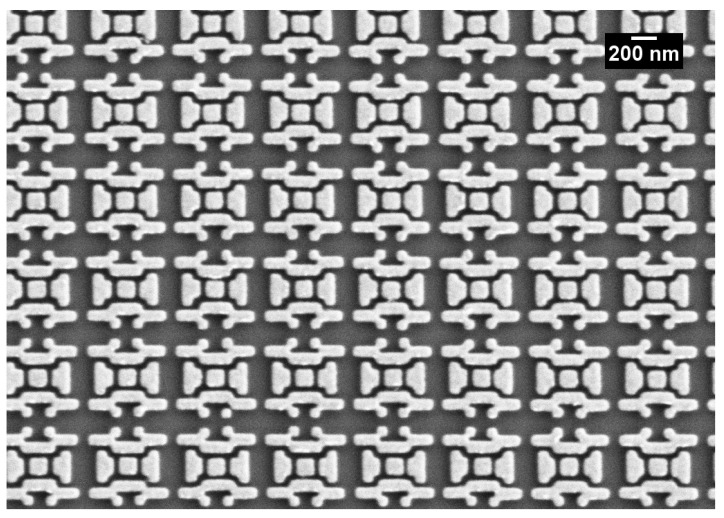
The fabrication result of the optimized SERS substrate captured by a scanning electron microscope (SEM). The unit cell was fabricated properly after the optimization of the fabrication parameters.

**Figure 6 nanomaterials-11-02905-f006:**
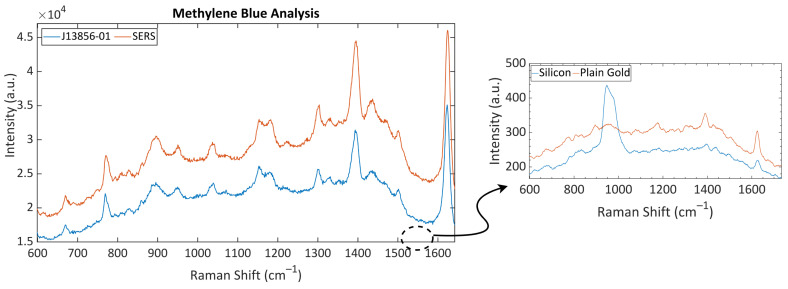
Raman spectra of methylene blue (MB) collected from the silicon, the plain gold, the industrial, and the optimized SERS substrates. The characteristic peaks of MB were detected in the spectrum collected from the optimized SERS substrate with a good SNR, comparable to the industrial SERS substrate J13856-01.

**Table 1 nanomaterials-11-02905-t001:** The obtained Raman spectra and attributed chemical structures (s, strong; m, medium; w, weak peak intensity). Each of the spectra and peaks were measured 30 times and were reproducible. The reproducibility of the spectra precludes any potential destructive effects of thermal drift due to enhanced localization.

Silicon (cm^−1^)	Plain Gold (cm^−1^)	SERS (cm^−1^)	Band Assignment [33]
-	676 (w)	683 (w)	Out-of-plane bending of C–H
-	773 (w)	774 (m)	In-plane bending of C–H
-	895 (w)	896 (m)	In-plane bending of C–H
-	-	1033 (w)	In-plane bending of C–H
-	1178 (w)	1170 (m)	Stretching of C–N
-	-	1299 (m)	In-plane ring deformation of C–H
-	-	1331 (m)	In-plane ring deformation of C–H
-	1389 (m)	1389 (s)	Symmetrical stretching of C–N
-	1427 (w)	1431 (m)	Asymmetrical stretching of C–N
1626 (w)	1625 (m)	1623 (s)	Ring stretching of C–C

**Table 2 nanomaterials-11-02905-t002:** Recent studies focused on the inverse design and topology optimization of different photonic components using different methods such as deep neural networks (DNNs), the genetic algorithm (GA), and the generative adversarial network (GAN).

Ref.	Method	Geometry	Figure of Merit	Wavelength
[38]	DNN	Core–Shell NP	Loss Function	-
[39]	GA	Metasurface	Reflection	600 nm
[45]	GA	Metasurface	Transmission	650–720 nm
[40]	GA	Metasurface	Polarization & Scattering	1.5 μm
[41]	GA	Metasurface	Transmission	16.9–44.7 mm
[42]	GA	Metasurface	Absorption	-
[43]	GAN	Metasurface	Transmittance	27.2 mm
[44]	GAN	Metasurface	Backpropagation efficiency	580 & 1550 nm

## Data Availability

The genetic algorithm script and a Lumerical design sample can be found at https://github.com/bebrem/sersgaoptimization, accessed on 20 October 2021.

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
