# Peer review of "Genetic Algorithm-Driven Surface-Enhanced Raman Spectroscopy Substrate Optimization"

_nanomaterials, 2021, doi:10.3390/nano11112905_

Round 1

Reviewer 1 Report

The authors make use of up-to-date computational optimization and high accuracy technology to fabricate a SERS-active substrate with improved sensitivity. The length of the paper is not justified, mainly because the main part is focused on the computational details of the GA-based optimization process in an unbalanced way as compared to the experimental input. For instance, why aren't the authors comparing the GA-optimized SERS substrate's performance by using commercial substrates or different GA-models? The comparison of the EF obtained for the detection of a common analyte when using the pattern or silicon or a simple gold film is not sufficient proof that the SERS-substrate was optimized. This process should include several patterns or comparative results with commercial substrates already characterized in literature in order to justify the effort of using GA-driven models in the first place. How is the reader convinced about the benefit of investing time and expertise resources in such a particular methodology? Please revise the discussion section because is too short and reduce the unnecessary computational details and afterwards reshape the conclusions of this study.

Author Response

Best regards

Reviewer 2 Report

This work presents a genetic algorithm (GA)-based SERS substrate optimization method for reconfigurable and programmable photonic SERS sensors for strong electric field localization over large areas. The authors analyze in detail the parameters of the GA and the tuning of the SERS substrate optimization. The authors also used experiments to verify the validation model results. This is a very valuable work, but I do not think it is particularly suitable for Nanomaterials and I would rather suggest the authors to submit the article to Sensors at MDPI. however, this is up to the editor's discretion. I think this article could meet the requirements for publication with revisions.

  1. Was the SERS substrate prepared using a method from the literature? If so, appropriate citations are needed.
  2. The discussion of data on Crossover Function and Elitism & Mutation Ratio does not give proper citations.
  3. Why did the authors choose this special structured nano-substrate?
  4. I don't think Table 1 is very necessary.

Author Response

Best regards

Reviewer 3 Report

Efficient and reproducible substrates are of basic importance for different applications of very sensitive SERS technique. Therefore, various substrates are intensively studied in many laboratories. The reviewed paper reports on application of a genetic algorithm for SERS substrate optimization. Subsequently, the best optimized nanostructure was fabricated to perform an experimental Raman test and estimate the SERS enhancement factor. In my opinion this is a valuable research which showed that the genetic algorithm can be used for fabrication of sensitive and reproducible substrates. It is important that such SERS substrates can be produced in a controllable way.  Previously, the genetic algorithm  was applied for optimizing some optical devices. Now, it is shown that it is also useful for design of efficient and controllable SERS substrates. I think that the paper can be accepted for publication in Nanomaterials, nevertheless I have some remarks and suggestions.

  • Page 2, lines 34-35. We read about “recent studies” which showed a large EF but no citation is given.
  • Page 4, line 123. It should be “… was represented with a 10x10 matrix.”
  • Page 5, line 166. The laser power of 18 mW is rather large, therefore it is possible that samples of the methylene blue were overheated. What was the area of the SERS chip on which the laser beam was focused?
  • Page 5. How it is defined the Average Distance Between Individuals?
  • Parameters of the genetic algorithm have been carefully tuned to find their optimal values for best SERS nanostructures. Each step of optimization is thoroughly described and illustrated in Figures 2-7. I appreciate this detailed description and believe that the optimization results are reliable. However, only the best optimized SERS substrate was fabricated and analyzed experimentally by Raman spectroscopy. The enhancement factor of this substrate was estimated in comparison with references which were the silicon and plane gold substrates; remarkable enhancement factors 8.8x106 and 6.3x106 have been obtained, respectively. However, nothing is known about improvement of the enhancement factor due to the substrate optimization, e.g. in comparison with the initial population. I think that some individuals of the initial population should also exhibit quite good enhancement factors. It is obvious that a significant improvement of the enhancement factors, determined experimentally on real SERS substrates, would give evidence that the genetic algorithm works efficiently. As a matter of fact, Figure 8 shows how the optimization procedure improves the effciency of substrates but these are not experimental data. I am aware that it would be difficult and time consuming to prepare some substrates from the initial population and to determine their properties experimentally, nevertheless I think it would be worthwhile to mention about this problem at least. May be the Authors could say anything interesting about it?   
  • The paper is well written, however I would like to ask for careful reading the manuscript once more. It is necessary to correct some sentences form the linguistic point of view. Moreover, in my opinion some sentences are so complicated that they are difficult for understanding.

Author Response

Best regards

Round 2

Reviewer 1 Report

I appreciate the fact that the authors took the time to test, in parallel, by using a commercial SERS substrate and to compare their results .

After verifying the modified manuscript, I however do not agree with the authors statement below fig 6, "The optimized SERS substrate has a powerful performance comparable to the industrial SERS chip. In addition, it is observed that the deformation peaks cannot be determined in the industrial chip or can be determined with very low contrasts."

I really don't understand their opinion since the two compared spectra are genuinely identical. It is my humble opinion that these are one and the same, at least from the quality of the figure. I cannot accept this statement. Either rephrase, either justify it by facts, zoom in in the region of those deformation modes and make actually visible any spectral difference. Or please provide an enlarged figure showing all spectra with clear details.

Author Response

Kind regards.
